# Comparison of intravenous magnesium sulphate and lidocaine for attenuation of cardiovascular response to laryngoscopy and endotracheal intubation in elective surgical patients at Zewditu Memorial Hospital Addis Ababa, Ethiopia

Abebaw Misganaw[1]*, Mulualem Sitote[2], Suliman Jemal[2], Eyayalem Melese[2], Metages Hune[3], Fetene Seyoum[4], Alekaw Sema[5], Dagim Bimrew[4]

1 Department of Anesthesia, School of Medicine, Debre Markos University, Debre Markos, Ethiopia, 2 School of Anesthesia, College of Health Sciences, Addis Ababa University, Addis Ababa, Ethiopia, 3 Department of Anesthesia, College of Medicine and Health Sciences, Debre Tabor University, Debre Tabor, Ethiopia, 4 Department of Anesthesia, College of Medicine and Health Sciences, Dire Dawa University, Dire Dawa, Ethiopia, 5 Department of Midwifery, College of Medicine and Health Sciences, Dire Dawa University, Dire Dawa, Ethiopia

* abebawmisganaw@gmail.com

## Abstract

### Background

Laryngoscopy and endotracheal intubation are essential components of general anesthesia. But it is always associated with side effects called reflex cardiovascular responses. Many methods have been identified to attenuate these responses like intravenous lidocaine, deep inhalational anesthesia, vasodilators, intravenous magnesium sulphate even though therapeutic superiority remains understudied.

### Methods

An institutional-based cohort study on 112 adult patients aged between 18–60 years was applied. 37 patients in the non-exposed group (Group N), 37 in the lidocaine group (Group L), and 38 in magnesium sulphate (Group M) were included. The hemodynamic parameters like heart rate, systolic, diastolic and mean arterial blood pressure at various time points up to 7 minutes post-intubation were recorded and the effect of both drugs to reduce hemodynamic responses was compared. Parametric data were analyzed using ANOVA and nonparametric data using the Kuruska-Wallis H rank test. P-value < 0.05 considered statistically significant.

### Results

In all three groups, there was a statistically significant rise in heart rate and blood pressure from baseline. There was a statistically significant difference in mean heart rate throughout

**Data Availability Statement:** The minimal data set supporting the conclusions of this article is available from the corresponding author and also uploaded as 'supporting information'.

**Funding:** The author(s) received no specific funding for this work.

**Competing interests:** The authors have declared that no competing interests exist.

**Abbreviations:** ASA, American Society of Anesthesiologists; BSc, Bachelor of Science; BP, Blood Pressure; CABG, Coronary Artery Bypass Grafting; DBP, Diastolic pressure; DRERC, Departmental Research and Ethics Review Committee; ETI, Endotracheal intubation; GA, General anesthesia; Group L, lidocaine group; Group M, magnesium sulphate group; Group N, non-exposed group; HR, Heart Rate; IV, intravenous; MAP, Mean arterial blood pressure; MgSO4, Magnesium sulphate; MSc, Master of Science; NMDA, *N* methyl-d-aspartate; OR, Operation room; PACU, post anesthesia care unit; SBP, Systolic blood pressure; SPSS, Statistical Package for Social Science.

study minutes among the groups (p<0.001). However, there was no statistically significant difference in mean heart rate between Groups M and L at all post-intubation time intervals. In blood pressure at all three parameters there was statistically significant difference among groups at all-time points except no difference at 7th minutes in DBP. There was significantly lower blood pressure in group M compared to both groups.

## Conclusion

In conclusion, prophylactic administration of magnesium sulphate and lidocaine was effective in attenuating hemodynamic responses to the stress effect of laryngoscopy and intubation. But based on our finding prophylaxis of magnesium sulphate is associated with a more favorable hemodynamic response.

## Background

Endotracheal intubation is an essential component of general anesthesia. It serves in the maintenance of patency of upper airway, proper ventilation, reduction in the risk of aspiration, and delivery of the inhalational anesthetic agents to the patients through breathing circuits [1]. Laryngoscopy and tracheal intubation are considered the most critical events during induction of general anesthesia which stimulate somatic and visceral nociceptive afferents fibers which induce reflex sympato-adrenal responses associated with enhanced neuronal activity in the cervical sympathetic efferent fibers [2].

Sympathetic stimulation from laryngoscopy and endotracheal intubation causes a significant increase in the plasma concentration of catecholamines (adrenaline and noradrenaline) [3, 4] that can provoke left ventricular failure, renal failure, surgical bleeding, cerebral hemorrhage and myocardial ischemia in anesthetized patients. The mechanism of this may be that, vasoconstriction, increased myocardial work, a demand for increased coronary flow, narrowed coronary arteries cannot accommodate the increased flow, and parts of the myocardium may receive insufficient oxygen [5–7].

The rise in blood pressure and heart rate is usually variable and unpredictable. The reflex tachycardia and hypertension effects of laryngoscopy are greater than of tracheal intubation. Once the endotracheal tube is in position, and the laryngoscope withdraws; hypertension, tachycardia, and disturbing dysrhythmia subside but tended to persist for up to 3–10 minutes. Hypertensive patients are more prone to exaggerated cardiovascular response to laryngoscopy and tracheal intubation than normotensive patients [2, 8–13].

Change in mean MAP during laryngoscopy and endotracheal intubation from baseline ranges from 23 to 52 mmHg [5, 14, 15]. Mean heart rate change from baseline after laryngoscopy & intubation ranges from 20 to 31 beat/minute [16, 17].

Both Magnesium sulphate and Lidocaine showed attenuation to presser response to laryngoscopy and endotracheal intubation with a different success rate in previous studies [9–11, 13, 18–24]. Many methods have been identified to attenuate these responses including topical anesthesia of oropharynx, laryngotracheal instillation of lidocaine before intubation, intravenous lidocaine, deep inhalational anesthesia, narcotics, vasodilators, intravenous magnesium sulphate, adrenergic and calcium blockers even though these techniques have drawbacks [25, 26]. Therefore this study aimed to compare the effect of intravenous lidocaine and magnesium sulphate on the attenuation of cardiovascular responses after laryngoscopy and endotracheal intubation in elective surgical patients.

## Materials and methods

### Study setting and period

The study was conducted in Zewditu Memorial Hospital. It provides service to an estimated above 800,000 people annually in the different departments who are referred from different zone of the city as well as all over the country. It has five major operation rooms and two post-anesthesia care unite (PACU). The hospital provides surgical services for about 10,000 patients annually. The research was conducted from November 7–2018 to March 7–2019. Patients who were induced with thiopental as non-exposed, premedicated with either iv lidocaine or magnesium sulphate, age 18 up to 60 years, and ASA I &II were included in the study. Patients on beta/Calcium channel blockers, premedicated with anticholinergic, hypertensive patients whose blood pressure > 140/90 mmHg, hypotensive patients whose systolic blood pressure <90 mmHg, hypothyroidism or hyperthyroidism, difficult intubation were excluded.

### Sample size and sampling procedure

The sample size was calculated using the previous study done in Iran in 2013 [18] by taking mean HR, SBP, DBP, MAP, and the largest sample size was taken using the comparison of two mean with equal sample size formula and using 80% power,α = 0.05. By adding a 10% non-response rate, the final sample size was 112. Study participants were selected using systematic random sampling technique using skip interval from the daily operation in the operation room (OR) in those patients induced with thiopental with or without ether lidocaine or magnesium sulphate premedication were used as a sampling frame.

A situational analysis done for one month, on average 3 patients per day or 60 patients per month were undergone surgery using thiopental as an induction agent with or without study drugs in Zewditu Memorial Hospital.

- ✓ Thus, 240 patients were operated per the study period (4 months). The sampling interval; K was determined using the formula: K = N/n; where, n = total sample size, N = population per 4 months. K = 240/112 ≈ 2

- ✓ Therefore, the sampling interval was two and the first study participant (random start) was selected using a lottery method from those induced with thiopental with or without study drugs who fulfill selection criteria.

- ✓ Then, every second case who induced with thiopental with or without study drugs was included in study groups until the required sample size was filled during the study period.

### Data collection procedures

Data was collected by a pretested structured questionnaire which enabled to take all necessary information from the chart of the patients and measured vital signs displayed on the monitoring screen. The study drugs lidocaine and magnesium sulphate in the study hospital are standardized or used routinely as part of preoperative care. We didn't assign patients for research purpose we just observed the anesthetists' discretion of treatment and some anesthetists used either lidocaine or magnesium sulphate for attenuation of hemodynamic reflex secondary to laryngoscopy & tracheal intubation but some anesthetists intubated patients without using both lidocaine and magnesium sulphate. Patients who refused to take part in the study were excluded in the study but they were received similar care with study participants.

On arrival of the patients to the operative theater the routine hospital monitoring protocol, HR, noninvasive blood pressure, and SPO2 were applied and after a room anesthetist decided

to induce with thiopental, data collectors took verbal informed consent of patients. After pre-oxygenation of patients with 100% oxygen for 3 minutes, anesthetists induced patients with thiopental 5 mg/kg and suxamethonium 2mg/kg with or without pretreatment of either magnesium or lidocaine and tramadol 100 mg IV for all patients. Lidocaine group (Group L) received 1.5 mg/kg lidocaine2%, magnesium sulphate group (Group M) received 30 mg /kg magnesium sulphate 50%, and Group N patients who induced with thiopental without taking either premedication drugs. Lidocaine and magnesium sulphate were given before 5 minutes of induction of anesthesia. Magnesium sulphate was injected slowly within 5 minutes. In our study area, anesthetists who used either lidocaine or magnesium had the same practice regarding the dose. Socio-demographic data like the patient's age, sex, and ASA physical status, BMI, associated coexisting illness were recorded from the chart. Mean arterial pressure, systolic blood pressure, diastolic blood pressure, Heart rate and SpO2 were recorded as the baseline (i.e., before starting of administration of magnesium sulfate or lidocaine for exposed or thiopental for non-exposed), 1 minute after injection of study drug, immediately after intubation (i.e., within 30 seconds after intubation), at $2^{nd}$ minutes post-intubation, at $5^{th}$ minutes post-intubation and at $7^{th}$ minutes post-intubation.

Hypertension was considered when the BP values of SBP>140 or DBP > 90 mmHg. Hypotension was considered when BP values of SBP < 90 mmHg. Tachycardia was considered when HR > 100 bpm. Bradycardia was considered when HR value lower than 50 bpm.

## Data quality control and assurance

Data was collected using a pretested structured questionnaire which enabled to review the chart records and measured vital signs displayed on the monitoring screen prepared in English addressing the objective of the study. The pretest was done on 5% of the sample size at Minilik II referral hospital. Data collectors were Anesthetists who are familiar with recording perioperative data. Data collectors and supervisors were trained on each item included in the study tools, objective, relevant of study, right of respondents. During data collection, regular supervision and follow up was made. The investigator cross-checked for completeness and consistency of data on a daily basis.

## Data processing and analysis

The statistical analyses were performed using SPSS 20 software. The data were tested for normality using the Shapiro–Wilk normality test and histogram. The homogeneity of variance also checked by Levene's test and Mauchly's test for sphericity. One way Analysis of variance (ANOVA) and repeated measure ANOVA was used for normally distributed continuous data. Kruskal–Wallis H test was used for non-normally distributed data. If the ANOVAs test was significant, then the Tukey post hoc test was used to compare one group with the others. Categorical data were analyzed using the Pearson Chi-squared test. Continuous variables were expressed as a mean & standard deviation (SD) and Median (Q1-Q3). Categorical variables were summarized by percentages. P-value < 0.05 considered statistically significant.

## Ethics approval and consent to participate

Before the data collection, ethical clearance was obtained from the Departmental Research and Ethics Review Committee of the Department of anesthesia, School of Medicine, College of Health Sciences of Addis Abba University. The purpose and importance of the study were explained to study participants and the director of the hospital. We believed the study is free of any risk since the study was an observational cohort and also the study-drugs were standardized, therefore, we used verbal consent. There was no issue to obtain a written consent and the

IRB approved the use of verbal consent. The obtained oral consent recorded by ticking on 'yes' if the participant agreed to participate in the study and 'no' if not agreed. All findings were kept confidential. The name and addresses of the participants were not recorded in the questionnaire. Furthermore, all the basic principles of human research ethics (respect for a person, beneficence, voluntary participation, confidentiality, and justice) were valued.

## Results

### Demographic and clinical characteristics of the patients

One hundred twelve patients were analyzed in this study. Thirty-seven patients in the non-exposed group (Group N), Thirty-seven in the lidocaine group (Group L), and Thirty-eight in magnesium sulphate (Group M) were included in this study. There was no significant difference among the three groups concerning age, gender, BMI, diagnosis, ASA physical status, type & MAC% of inhalational agents, surgery starting time, and maintenance muscle relaxant (p-value > 0.05) as showed in Table 1.

**Comparison of mean heart rate at different time points among magnesium sulphate, lidocaine, and non-exposed groups.** At baseline, Mean Heart Rate (HR) among the groups did not show a significant difference statistically (p = 0.436). The one way ANOVA analysis showed that there was a statistically significant difference in mean heart rate throughout study minutes among the groups (p<0.001). And the post hoc analysis showed that mean HR was higher in Group N with statistically significant value compared to both groups (p<0.001). However, there was no statistically significant difference in mean heart rate between Groups M and L at the immediate, 2nd, 5th, and 7th post-intubation time intervals(p = 0.324,0.222, 0.356,0.737) respectively (Table 2). Regarding within-the group comparison, there was a statistically significant rise in mean heart rate from baseline in all study groups throughout study minutes (p< 0.05) (Fig 1).

**Table 1. Demographic data and anesthetic characteristics of patients in Zewditu Memorial Hospital, Addis Ababa in 2018/2019.**

| Characteristics | | Group M | Group L | Group N | P-value |
|---|---|---|---|---|---|
| Age (years) | Mean ±SD | 32.32±7.19 | 36.05±7.69 | 34.68±8.34 | 0.112 |
| Sex (F/M) | Female (%) | 84.2 | 78.4 | 89.2 | 0.448 |
| | Male (%) | 15.8 | 21.6 | 10.8 | |
| ASA status | ASA I (%) | 100 | 100 | 100 | - |
| BMI | Median (Q3-Q1) | 24 (25–23) | 24 (25–23) | 24 (25–23) | 0.660 |
| Diagnosis | Goiter (%) | 47.4 | 40.5 | 51.4 | 0.843 |
| | Cholilethiasis (%) | 28.9 | 35.1 | 32.4 | |
| | Neurosurgery (%) | 23.7 | 24.4 | 16.2 | |
| Maintenance inhalational agents | Isoflurane (%) | 81.6 | 67.6 | 75.7 | 0.372 |
| | Halothane (%) | 18.4 | 32.4 | 24.3 | |
| MAC of inhalational agent | MAC1% (%) | 18.4 | 27 | 10.8 | 0.103 |
| | MAC1.5%(%) | 78.9 | 67.6 | 73.0 | |
| | MAC2%(%) | 2.7 | 5.4 | 16.2 | |
| Maintenance muscle relaxant within 7 minutes | Pancuronium (%) | 28.9 | 10.8 | 13.5 | 0.286 |
| | Vecuronium (%) | 10.5 | 10.8 | 10.8 | |
| | No muscle relaxant within 7 minutes (%) | 60.6 | 78.4 | 75.7 | |
| Is surgery started within 7 minutes of intubation? | Yes (%) | 15.8 | 21.6 | 16.2 | 0.765 |
| | No (%) | 84.2 | 78.4 | 83.8 | |

Data are analyzed by ANOVA, Kruskal Wallis, and Chi-Square Test, BMI-Body mass index, MAC- minimum alveolar concentration.

**Table 2. Comparison of mean heart rate among the groups and between groups in Zewditu Memorial Hospital, Addis Ababa, 2018/2019.**

| Time interval | Group M | Group L | Group N | Significance of the difference among the groups | Comparison between Group M & Group L |
|---|---|---|---|---|---|
| | Mean ± SD | Mean ± SD | Mean ± SD | P-value | P-value |
| Baseline | 80.97±6.28 | 80.24±4.85 | 82.16±7.84 | 0.436 | 0.876 |
| Immediate Post Intubation | 99.82 ±10.86 | 96.35±9.25 | 120±11.04 | < .001*, # | 0.324 |
| 2min post Intubation | 94.47 ±11.37 | 90.49±9.64 | 112.22±9.93 | < .001*, # | 0.222 |
| 5min post Intubation | 88.13±9.58 | 84.84±8.28 | 104.46 ±12.70 | < .001*, # | 0.356 |
| 7min post Intubation | 85.79±8.48 | 84.03±8.69 | 100.65 ±12.95 | < .001*, # | 0.737 |

* $p<0.05$ compared group N with M

# P value $<0.05$ compared group N with L (ANOVA, Tukey test).

**Comparison of mean SBP at different time points among magnesium sulphate, lidocaine, and non-exposed groups.** There were no statistically significant intergroup differences in baseline SBP among groups (p = 0.655). There was a statistically significant difference in mean SBP among groups at the all-time points. Post hoc analysis showed significant lower mean SBP at immediate, 2nd, and 5th minutes post-intubation in group M compared to Group L (p <0.001, p = 0.001, p = 0.029), respectively. And there was also a statistically significant decrement in mean SBP when group M compared to group N at the immediate, 2nd, 5th (p<0.001), and 7th minute post-intubation (p = 0.006) but there was no significant difference between two treatment groups at 7th minute. There was also a statistically significantly lower mean SBP in group L compared with group N at all-time points except at 7th minute (Table 3).

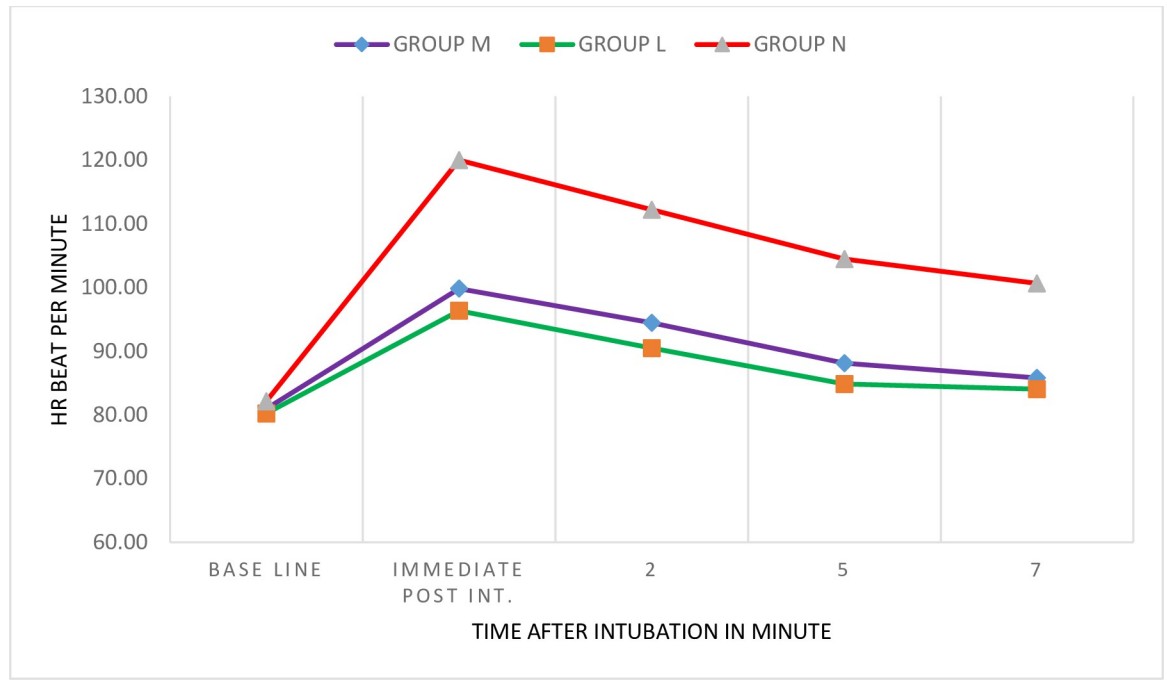

**Fig 1. Within-the group change in heart rate at different time intervals in Zewditu Memorial Hospital, Addis Ababa, 2018/2019.**

**Table 3. Comparison of mean SBP among the groups and between groups at different time intervals in Zewditu Memorial Hospital, Addis Ababa, 2018/2019.**

| Time Interval | Group M | Group L | Group N | Significance of the difference among the groups | Effect size |
|---|---|---|---|---|---|
| | Mean ± SD | Mean ± SD | Mean ± SD | P-value | η2 |
| Baseline | 126.66±7.48 | 126.62±6.37 | 127.78±4.14 | 0.655 | - |
| Immediate Post intubation | 142.71±10.14 | 155.27±12.62 | 179.27±17.82 | < .001*, #, + | 0.55 |
| 2min post Intubation | 129.58±10.96 | 141.54±11.47 | 152.95±17.98 | < .001*, #, + | 0.33 |
| 5min post Intubation | 119.26±7.94 | 125.68±10.36 | 134.22±13.18 | < .001*, #, + | 0.25 |
| 7min post Intubation | 115.71±7.81 | 118.35±10.08 | 122.49±9.81 | 0.003* | 0.08 |

* p<0.05 compared group N with M

\# P value <0.05 compared group N with L

+p<0.05 compared group L with group M (ANOVA, Tukey test).

Regarding within-the group comparison there was a significant rise in SBP in group M at the immediate post-intubation time only (p < 0.001) and return to baseline at the 2nd minute of intubation. In the lidocaine group, there was a significant rise in SBP at the immediate and at 2nd minute post-intubation (*p*< 0.001) and return to baseline at 5th minute of intubation (*p* = 0.643) whereas in non-exposed group significant rise in SBP continued till the fifth post-intubation minute (Fig 2).

**Comparison of mean DBP at different time points among magnesium sulphate, lidocaine, and non-exposed groups.** The baseline DBP was comparable among the three groups (p = 0.194) and there was a statistically significant difference among the groups at immediate, 2nd, and 5th minutes post-intubation intervals but not at 7th minutes post-intubation. Post hoc analysis showed that group M has significantly lower mean DBP at immediate, 2nd, and 5th minutes post-intubation compared to Group N (p < 0.001). There was also a statistically

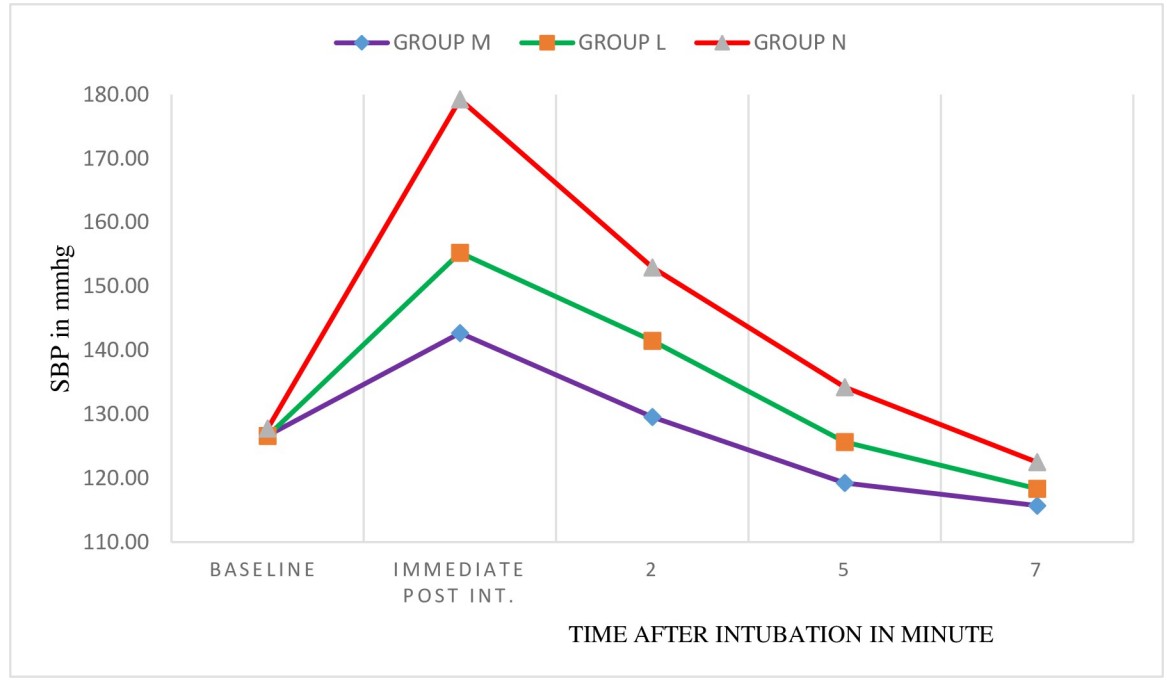

**Fig 2. Within-the group change in SBP at different time intervals in Zewditu Memorial Hospital, Addis Ababa, 2018/2019.**

**Table 4. Comparison of mean DBP among the groups and between groups at different time intervals in Zewditu Memorial Hospital, Addis Ababa, 2018/2019.**

| Time Interval | Group M | Group L | Group N | Significance of the difference among the groups | Effect size |
|---|---|---|---|---|---|
| | Mean ± SD | Mean ± SD | Mean ± SD | P-value | η2 |
| Baseline | 77.16±5.23 | 74.76±5.61 | 75.73±6.28 | 0.194 | - |
| Immediate Post Intubation | 91.29±8.29 | 97.70±10.28 | 112.81±11.40 | < .001*, #, + | 0.45 |
| 2min post Intubation | 81.08±8.07 | 87.49±9.42 | 97.73 ±12.36 | < .001*, #, + | 0.32 |
| 5min post Intubation | 72.87±7.44 | 77.11±7.79 | 84.59±11.96 | < .001*, # | 0.22 |
| 7min post Intubation | 71.00±6.70 | 73.14±8.83 | 74.62.±6.98 | 0. 183 | - |

* p<0.05 compared group N with M

#P value <0.05 compared group N with L

+p<0.05 compared group L with group M (ANOVA, Tukey test).

significant lower mean DBP in Group L at the immediate, 2nd, and 5th minutes post-intubation compared to Group N (p < 0.001, < 0.001, = 0.002); respectively. There was no statistically significant difference in mean DBP between Group M and Group L except at immediate and 2nd minute post-intubation periods (p = 0.018, 0.019) respectively (Table 4). Regarding within-the group comparison, there was a significant rise in DBP in group M only at immediate and at 2nd minute of intubation with (p < 0.001, = 0.010) respectively. Similarly in the lidocaine group, there was a significant rise in DBP at immediate and at 2nd minute of post-intubation (p< 0.001) whereas in the non-exposed group significant rise in DBP continued till the fifth minute (p< 0.001).

**Comparison of mean MAP at different time points among magnesium sulphate, lidocaine, and non-exposed groups.** Regarding baseline-MAP, groups were matched (p = 0.548). There was a statistically significant difference in mean MAP among all groups at all-time points (Table 5). Post hoc analysis showed significant lower mean MAP at immediate, 2nd, 5th minutes post-intubation in group M compared to Group L (p = 0.002, p = 0.001, p = 0.023 respectively). Group M compared to group N, mean MAP was significantly lower at immediate, 2nd, 5th and at 7th minutes post-intubation intervals (p<0.001, <0.001, <0.001, = 0.011) respectively. Also Group L has significantly lower mean MAP at immediate, 2nd, and 5th minutes post-intubation intervals compared to Group N (p < 0.001, < 0.001, = 0.001 respectively). At 7th minute there was no statistically significant difference in MAP between group L and N (p = 0.310) (Table 5). Regarding within-the group comparison, there was a significant rise in MAP in group M at immediate and 2nd minute of intubation with (p < 0.001, = 0.041 respectively). Similarly in the lidocaine group, there was a significant rise in MAP at immediate

**Table 5. Comparison of mean MAP among the groups and between groups at different time intervals in Zewditu Memorial Hospital, Addis Ababa, 2018/2019.**

| Time Interval | Group M | Group L | Group N | Significance of the difference among the groups | Effect size |
|---|---|---|---|---|---|
| | Mean ± SD | Mean ± SD | Mean ± SD | P-value | η2 |
| Baseline | 93.71±4.26 | 92.51±5.54 | 92.78±4.97 | 0.548 | - |
| Immediate Post Intubation | 108.29±8.40 | 116.95±10.29 | 134.65±13.36 | < .001*, #, + | 0.51 |
| 2min post Intubation | 96.47±7.56 | 105.57±8.97 | 116.03±13.41 | < .001*, #, + | 0.38 |
| 5min post Intubation | 87.87±7.30 | 93.35±7.56 | 100.95±11.20 | < .001*, #, + | 0.27 |
| 7min post Intubation | 85.53±6.88 | 88.30±8.40 | 91.09±9.05 | 0.015* | 0.07 |

* p<0.05 compared group N with M

#P value <0.05 compared group N with L

+p<0.05 compared group L with group M (ANOVA, Tukey test).

and $2^{nd}$ minute of post-intubation (p< 0.001) whereas in the non-exposed group significant rise in MAP continued till the fifth minute (p< 0.001).

## Discussion

In our prospective cohort study demographic data, anesthetic characteristics of patients, and baseline hemodynamic variables (SBP, DBP, MAP, and HR) were comparable in both groups.

### Heart rate

In this study the peak mean heart rates occurred at the immediate post-intubation time, which was 99.82±10.86, 96.35±9.25, 120.0±11.04 in group M, L N respectively with (p <0.001). Our study showed that there was a statistically significant difference in mean heart rate at all post-intubation time intervals among the groups (p<0.001). The mean heart rate in Group N was significantly higher compared with Group L and M at all post-intubation time intervals. However, there was no statistically significant difference in mean heart rate between Groups M and L at all post-intubation time intervals. In this study, a statistically significant increase in mean heart rate from baseline was observed in Group M, Group L, and Group N at all post-intubation time intervals. Bandey S. et. al in 2016 noted Similar findings [11].

Our study is also consistent with other studies done by Bhalerao NS et al (2017) reported that the difference in HR was not statistically significant between magnesium sulphate and lidocaine groups throughout the study period. In contrary to our study, they found no statistically significant change in mean heart rate from baseline in both groups regarding within-the group comparison [10]. The possible difference from our study may be the variation in the drug used for induction (propofol) and the dose of pretreatment (magnesium sulphate: 50mg/kg and lidocaine: 2mg/kg but in our study anesthetists used 30mg/kg and 1.5mg/kg respectively).

In contrary to our finding, Waseem M et al (2011) showed that there was a statistically significant difference between magnesium sulphate and lidocaine groups in attenuating increment of heart rate. A higher percentage of patients in the magnesium group (25.6%) than the lidocaine group (12.35%) had heart rate increment from baseline by >25% from baseline [27]. The possible explanation for this different result might be due to a low dose of magnesium sulphate (10mg/kg) but in our study anesthetists used 30mg/kg.

### Systolic blood pressure

Our study demonstrated a statistically significant difference in mean SBP among all groups at all time points. There was lower mean SBP at the immediate, $2^{nd}$, and $5^{th}$ minutes of post-intubation in group M compared to Group L (p <0.001, p = 0.001, p = 0.029 respectively) and compared to group N (p<0.001). Again there was significantly lower mean SBP in group L compared with group N at all-time points except at $7^{th}$ minute. At $7^{th}$ minute there was a significantly lower mean SBP in group M compared with group N ($p$ = 0.006) but there was no significant difference between the two treatment groups. On within-the group comparison there was a significant rise in SBP in group M at the immediate post-intubation time only (p < 0.001) and return to baseline at $2^{nd}$ minute of intubation. In the lidocaine group, there was a significant rise in SBP at the immediate post-intubation time and $2^{nd}$ minute post-intubation (p< 0.001) and there was no statistically significant difference at $5^{th}$ minute post-intubation time (p = 0.643) whereas in non-exposed group significant rise in SBP continued till $5^{th}$ minute.

Our study is in line with a study done by Sachin Padmawar, et al (2016) reported that there was a significantly higher mean SBP in the lidocaine group as compared with the MgSO4

group at 1,3,5 minutes after intubation. Regarding within-the group comparison, SBP increased significantly from baseline. But it came to baseline within 5 minutes in the magnesium group, whereas in the lidocaine group did not come to baseline value within 5 minutes in our study these parameters came to baseline faster [13]. The possible reason might be in their study used low volume% of halothane for maintenance (0.4–0.6%).

Our study is also consistent with other study done by Nooraei N et al (2013) found a statistically significant difference in mean SBP between magnesium sulphate and lidocaine groups at 1st and 2nd minutes(p = 0.001,0.033)respectively with higher value in the lidocaine group [18].

In contrast to our study, done by Mendonca FT et al (2016) they compared the episodes of hypertension (increase in SBP >20%of baseline) after intubation and there was no statistical significance difference between magnesium and lidocaine groups. Three patients in the magnesium group (12%) had compared to one patient (4%) in the lidocaine group with (p >0.05) they found magnesium has similar results to lidocaine [23]. The possible difference with our finding might be due to variation in induction agent they used propofol and fentanyl pretreatment in our study not.

## Diastolic blood pressure

There was a statistically significant difference among all groups at immediate, 2nd, and 5th minutes post-intubation intervals but not at 7th minute of post-intubation. Group M has a statistically significantly lower mean DBP at immediate, 2nd, and 5th minutes post-intubation intervals compared to Group N (p < 0.001). Also Group L has significantly lower mean DBP at immediate post-intubation, 2nd and 5th minutes post-intubation intervals compared to Group N (p < 0.001, < 0.001, = 0.002). Group M compared to Group L has significantly lower value only at immediate and at 2nd minutes post-intubation periods with (p = 0.018, 0.019 respectively). DBP rose significantly from baseline in group M at immediate and 2nd minutes of intubation with (p < 0.001, = 0.010 respectively). Similarly in the lidocaine group, there was a significant rise in DBP at immediate and 2nd minutes post-intubation with (p< 0.001) whereas in the non-exposed group a significant rise in DBP continued till the fifth minute with (p< 0.001). Our findings are in line with two studies done in 2016 by Bandey S et al and Sachin Padmawar et al [11, 13].

In contrary to our findings, Nooraei N et al (2013) showed no statistically significant difference in mean DBP between magnesium sulphate and lidocaine groups throughout study minutes for five minutes [18]. The likely explanation for this inconsistency could be Nooraei N. et. al used fentanyl which is effective in attenuating hemodynamic response secondary to laryngoscopy and intubation [28].

## Mean arterial pressure

There was a statistically significant difference in mean MAP among all groups at all time points. There was also significant lower mean MAP at immediate, 2nd, and 5th minutes post-intubation in group M compared to Group L (p <0.001, p = 0.001, p = 0.029 respectively). Group M compared to group N, mean MAP was significantly lower at immediate, 2nd minutes, 5th minute and at 7th minutes post-intubation intervals (p<0.001,<0.001,<0.001, = 0.011 respectively). Group L has a significantly lower mean MAP at immediate, 2nd, and at 5th minutes post-intubation intervals compared to Group N (p < 0.001, < 0.001, = 0.001respectively). MAP rose significantly from baseline in group M at immediate and 2nd minutes of intubation with (p < 0.001, = 0.041 respectively). In the lidocaine group also there was a significant rise in MAP at immediate and 2nd minutes post-intubation with (p< 0.001) whereas in non-exposed group significant rise in MAP continued till the fifth minute with (p< 0.001). In line with our

study done by R Vallabha et al (2018) reported mean MAP in lidocaine and magnesium sulphate group respectively were 88.70+8.95 vs 79.83+7.34, 88.70+8.95 vs 79.83+7.34, 86.50+8.37 vs 78.03+7.10 at 1st, 3rd, and 5th minutes respectively (p < 0.001). They found a significantly lower MAP in magnesium sulphate compared with lidocaine group [29].

Inconsistent to our study done by G Kiraci, et al (2014) observed no significant difference in MAP at immediate, 2nd, 5th and 10th post-intubation minutes between magnesium, lidocaine, and control groups ($P > 0.05$) [30]. The possible controversy from our study might be the variation in the drug used for induction (propofol) and the dose of pretreatment (magnesium sulphate: 10mg/kg and lidocaine: 1mg/kg but in our study anesthetists used 30mg/kg and 1.5mg/kg respectively).

## Limitation

The current study has certain limitations such as inability to control over the confounding factors like type of inhalational agent for maintenance, MAC of inhalational agent, diagnosis and maintenance muscle relaxant in addition in our study setting the study drugs are based on the preference of anesthetists since the design is observational.

## Strength

We have tried to make comparable study groups by including patients who induced with same induction agent. We had no incomplete data with missing values, adequate sample size was attained on the planned schedule of time. So that the difference observed may be due to exposure factors.

## Relevance of the study

Based on our finding we recommend prophylactic IV magnesium sulphate and lidocaine to consider for attenuating hemodynamic response to laryngoscopy and endotracheal intubation. We also recommend additional randomized clinical trial.

## Conclusion

In conclusion, prophylactic administration of magnesium sulphate and lidocaine was effective in attenuating hemodynamic responses of laryngoscopy and endotracheal intubation. But based on our finding prophylaxis of magnesium sulphate is associated with a more favorable hemodynamic response.

## Supporting information

**S1 File.**
(SAV)

**S2 File.**
(DOCX)

## Acknowledgments

We would like to thank the anesthesia department of Addis Ababa University for all their support and constructive comments.

We would like also to extend our appreciation to Zewditu Memorial Hospital medical directorate, anesthesia staff, and data collectors for their assistance and cooperation in the completion of this study.

## Author Contributions

**Conceptualization:** Abebaw Misganaw, Mulualem Sitote, Suliman Jemal, Eyayalem Melese, Metages Hune, Fetene Seyoum, Alekaw Sema, Dagim Bimrew.

**Data curation:** Abebaw Misganaw, Suliman Jemal, Fetene Seyoum, Alekaw Sema, Dagim Bimrew.

**Formal analysis:** Abebaw Misganaw, Mulualem Sitote, Suliman Jemal, Eyayalem Melese, Metages Hune, Fetene Seyoum, Alekaw Sema, Dagim Bimrew.

**Funding acquisition:** Abebaw Misganaw, Mulualem Sitote, Suliman Jemal, Eyayalem Melese, Metages Hune, Fetene Seyoum, Alekaw Sema, Dagim Bimrew.

**Investigation:** Abebaw Misganaw, Mulualem Sitote, Suliman Jemal, Eyayalem Melese, Metages Hune, Fetene Seyoum, Alekaw Sema, Dagim Bimrew.

**Methodology:** Abebaw Misganaw, Mulualem Sitote, Suliman Jemal, Eyayalem Melese, Metages Hune, Fetene Seyoum, Alekaw Sema, Dagim Bimrew.

**Project administration:** Abebaw Misganaw, Mulualem Sitote, Suliman Jemal, Eyayalem Melese, Metages Hune, Fetene Seyoum, Alekaw Sema.

**Resources:** Abebaw Misganaw, Suliman Jemal, Eyayalem Melese, Metages Hune, Fetene Seyoum, Alekaw Sema, Dagim Bimrew.

**Software:** Abebaw Misganaw, Mulualem Sitote, Suliman Jemal, Eyayalem Melese, Metages Hune, Fetene Seyoum, Alekaw Sema, Dagim Bimrew.

**Supervision:** Abebaw Misganaw, Mulualem Sitote, Suliman Jemal, Eyayalem Melese, Metages Hune, Fetene Seyoum, Dagim Bimrew.

**Validation:** Abebaw Misganaw, Mulualem Sitote, Eyayalem Melese, Fetene Seyoum, Dagim Bimrew.

**Visualization:** Abebaw Misganaw, Mulualem Sitote, Suliman Jemal, Eyayalem Melese, Metages Hune, Fetene Seyoum, Alekaw Sema, Dagim Bimrew.

**Writing – original draft:** Abebaw Misganaw, Mulualem Sitote, Suliman Jemal, Eyayalem Melese, Metages Hune, Fetene Seyoum, Alekaw Sema, Dagim Bimrew.

**Writing – review & editing:** Abebaw Misganaw, Mulualem Sitote, Suliman Jemal, Eyayalem Melese, Metages Hune, Fetene Seyoum, Alekaw Sema, Dagim Bimrew.

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
