## [Decision Letter · Decision Letter 0]

25 Jan 2021

PONE-D-20-36884

Comparison of Intravenous Magnesium sulphate and Lidocaine for Attenuation of Cardiovascular Response to Laryngoscopy and Endotracheal Intubation in Elective Surgical Patients at Zewditu Memorial Hospital Addis Ababa, Ethiopia.

PLOS ONE

Dear Dr. Misganaw,

Thank you for submitting your manuscript to PLOS ONE. After careful consideration, we feel that it has merit but does not fully meet PLOS ONE’s publication criteria as it currently stands. Therefore, we invite you to submit a revised version of the manuscript that addresses the points raised during the review process.

All issues raised by expert reviewers are required.

We look forward to receiving your revised manuscript.

Kind regards,

Vincenzo Lionetti, M.D., PhD

Academic Editor

PLOS ONE

Journal Requirements:

2. Please ensure you have discussed any potential limitations of your study in the Discussion, including study design, sample size and/or potential confounders.

4. In the Methods, please clarify the following:

- Why written consent could not be obtained

- Whether the Institutional Review Board (IRB) approved use of oral consent

- How oral consent was documented

For more information, please see our guidelines for human subjects research: https://journals.plos.org/plosone/s/submission-guidelines#loc-human-subjects-research

7. Please amend either the title on the online submission form (via Edit Submission) or the title in the manuscript so that they are identical.

Reviewers' comments:

Reviewer's Responses to Questions

**Comments to the Author**

1. Is the manuscript technically sound, and do the data support the conclusions?

Reviewer #1: No

Reviewer #2: Partly

Reviewer #3: Yes

2. Has the statistical analysis been performed appropriately and rigorously? 

Reviewer #1: Yes

Reviewer #2: Yes

Reviewer #3: Yes

3. Have the authors made all data underlying the findings in their manuscript fully available?

Reviewer #1: Yes

Reviewer #2: Yes

Reviewer #3: Yes

4. Is the manuscript presented in an intelligible fashion and written in standard English?

Reviewer #1: No

Reviewer #2: Yes

Reviewer #3: No

5. Review Comments to the Author

Reviewer #1: PLOS

This study investigates whether the addition of lidocaine or magnesium attenuates increases in heart rate and blood pressure during intubation.

Major comments

- Abstract expected to be revised based on the suggestions below.

- The authors investigate and report both diastolic, mean and systemic blood pressure in great detail besides heart rate. Therefore the article contains a lot of data which makes it more difficult to grasp. I would strongly suggest the authors report heart rate and either systolic or mean arterial pressure in the main text. Other BP data can be provided as a supplement and referred to in the manuscript (e.g.”findings for diastolic and mean arterial pressure were similar, see Supplement XX”).

- A language revision would definitely improve readability of the manuscript.

- p6. Data was collected prospectively. Therefore it is unclear for me why only every second patient was chosen (in order to reach the four month time frame?). Why not every patient, increasing sample size or shortening the study period. Logistic reasons?

- p9. The number of patients in each group should be reported early in the results section (I could only find it in the abstract). Also, since group allocation was based on the anesthetists’ preference, it is quite surprising that group size was so comparable. Please explain.

- When describing results, I suggest to first describe the within-group changes followed by the between-group changes.

- p12. The results for SBP in the text and table 3 differ. Please check (e.g. p value at 5 minutes 0.029 or < 0.001).

- Please describe the incidence of bradycardia and hypotension in the results section and comment in the discussion.

- Discussion. Needs to be heavily restructured and rewritten. Actual numbers need not to be repeated in the discussion section but should be summarized with major findings highlighted. Try to compare the overall results of the study with the overall picture from other reports, not every parameter by itself.

- p22. I am afraid that the study does not support the conclusion that magnesium sulphate is ”better” than lignocaine. Firstly, as the authors touch upon in the discussion, the differences are undoubtedly dose-dependent. Secondly, the use of lidocaine, magnesium or nothing was dependent on the individual anesthetist. Thus other factors such as intubating skills, vaporizer technique etc. could influence the results. In my opinion, the only conclusion to be drawn from the data is rather descriptive: In this setting, with these doses, patients anesthetized by an anesthetist using lidocaine or magnesium had attenuated sympatic responses to intubation, compared to anesthetists using none, with a more pronounced effect in the group given magnesium.

Minor comments

p3. last line: I think increased catecholamines during intubation are a very rare cause of surgical bleeding and renal failure.

p4. The term mean MAP (several instances in the document) is confusing. I would suggest ”MAP”, and specify ”mean + SD” when appropriate.

last line ”at Zewditu ….. Ethiopia.” can be removed.

p5. Remove ””Kirkos sub city district 08”. Remove ”The hospital …July 2018”.

Please describe which difference was deemed significant for the sample size calculation. Also specify assumed means and SD’s based on the Iranian study.

p7. Please explain why opioids such as fentanyl were not used during induction, which to my knowledge would be standard in many parts of the world. Economic reasons? However all patients received tramadol. Is it cheaper than fentanyl?

Were drug doses per kg in actual body weight or ideal body weight.

Was there a specified time frame for injecting lidocaine or magnesium.

P9. Remove ”Following approval …of the hospital”.

Table 1 and elsewhere. It is sufficient to specify the hospital and study period once, this needs not to be repeated in all legends.

Table 1 MAC. At what time point was MAC assessed (it is highly dynamic at the start of anesthesia). Also, are the data presented in intervals? (e.g. below 1%, between 1 and 2%, above 2 %?)

P11. Fourth line. The P-values can be removed, they are already in Table 2.

P12. I would suggest ”SBP” instead of ”mean SBP”

Please describe in methods if/how the p-value was adjusted for multiple comparisons. Since many comparisons are made, a P value of 0.029 is not very convincing, or should it read 0.0029?

Please be consistent in using ”lidocaine” or ”lignocaine” in the document.

Reviewer #2: The topic is very interesting as clinical evidence to reduce cardiovascular response to laryngoscopy and endotracheal intubation especially in patients with CVS problems. You have used the terms lidocaine and lignocaine in different places. Make it uniform even though it is the same drug or justify.

You have said laryngoscopic stimulation has many cardiovascular responses but Does it always reflex cardiovascular response in all settings? What type of cardiovascular response do you expect?

in data collection procedure you have discussed that the study subjects have received the similar care. How do you give similar care for study subjects and who refused to participate in the study since you have no intervention?

Generally the study seems better if conducted with RCT design and your data collection procedure and subject recruitment also favors in that manner. so, it needs clarification on how it can be done to compare those drugs with observational study.

Reviewer #3: The paper covers somewhat an interesting topic and can be appealing for the readership of PLOS ONE. However - although the structure reads with ease - the discussion and conclusion part shall be reworded and rewritten minding to improve clarity and readability. In this regard, I am usually reluctant to question English writing, since I am not a native myself, even though in this case i strongly encourage authors to benefit from a professional/mother tongue English editor.

I do consider the paper publishable pending minor revision, which i resume in the list below for clarity:

1) English revision throughout the paper and in particular in the discussion section.

2) In the Data Collection Procedures section, is not clear at all. In particular explain more in details when, how, and the group description of pretreatment with respective drugs(i.e. Lidocaine, Magnesium, and control group).

3) Also in the Data Collection, it is not sufficiently explained the timing of the delay between drugs assumption and intubation.

4) Further discuss the pro and cons of the sampling method adopted.

6. PLOS authors have the option to publish the peer review history of their article (what does this mean?). If published, this will include your full peer review and any attached files.

Reviewer #1: **Yes: **Hans Bahlmann

Reviewer #2: No

Reviewer #3: **Yes: **Nora Terrasini; MD, PhD

---

## [Author Response · Author response to Decision Letter 0]

3 Mar 2021

First of all we would like to say thank you all reviewers for their detailed review. Then let us clear some questions and doubts raised by reviewers

Reviewer 1 (Hans Bahlmann) 

1. Why only every second patient was chosen? We used propoablity sampling method (SRS) and the calculation based on systematic random sampling is every second patient. 

2. P9. Group allocation is based on anesthetist’s preference? Our data collectors observed and collected data which fulfils the inclusion criteria then if the required number is found in any group before study period is finished they only collected the remain group. E.g. In Lidocaine group the required number reached within 3 months. 

3. P12. Table 3 contains p value of between the group and within the group p values. So at 5 min the p value between Group M and Group L is 0.029. and the p value among groups was <0.001

4. P22 our conclusion is based on our finding since the post-hoc analysis shows there is a statistically significance difference between lidocaine and magnesium groups. 

 Even though the nature of a study design unable to control confounding factors like vaporizers, intubation skill we exclude difficult intubations in the study. And there was no difference among the groups in the type and MAC of inhalational agents. 

Minor comment

1. P5. ‘Please describe which difference was deemed significant for the sample size calculation ‘? The Iranian study compared HR, SBP, DBP and MAP. We checked the difference between groups for all parameters and the difference was largest in MAP so we used this one for our sample calculation. 

2. P7. Why opioid like fentanyl was not used? In Ethiopia fentanyl was not available in the market and in study setting anesthetists used tramadol.

3. Drug dose was based on actual body weight.

4. Both lidocaine and magnesium were injected before 5 minutes of induction of anesthesia.

5. Table 1 MAC at what time point was MAC assessed? Since the study minutes was short (within 7 minutes) there was no great dynamics and we took the average value within 7 minutes of intubation. 

Reviewer2

1. Does cardiovascular reflex always happen in all setting? It may not necessarily in all setting but it happen at least in majority cases.

2. What type of cardiovascular response do you expect? The common ones are tachycardia, hypertension and arrhythmia are common responses to laryngoscopic stimulation. 

3. How do you give similar care for study subjects and who refused to participate? We just observed the event and if participants refused we only exclude from study participants otherwise the hospital anesthetists gave similar care for those included and excluded patients. Care was independent of being participant or not. 

4. Yes we agreed it is better if it was studied in RCT design, and how you can compare drugs in observational study? Even though it has limitation the hospital practice is somehow favorable for this study design from our previous observation and we tried our best to compare drugs with this study design. 

Reviewer 3(Nora Terrasini;MD, PHD)

1. Lidocaine and magnesium sulphate were given before 5 minutes of induction of anesthesia. Both drugs were given in IV push and magnesium sulphate was injected slowly within 5 minutes. After 5 minutes of pretreatment anesthetists induced anesthesia.

---

## [Decision Letter · Decision Letter 1]

22 Mar 2021

PONE-D-20-36884R1

Comparison of Intravenous Magnesium sulphate and Lidocaine for Attenuation of Cardiovascular Response to Laryngoscopy and Endotracheal Intubation in Elective Surgical Patients at Zewditu Memorial Hospital Addis Ababa, Ethiopia.

PLOS ONE

Dear Dr. Misganaw,

Thank you for submitting your manuscript to PLOS ONE. After careful consideration, we feel that it has merit but does not fully meet PLOS ONE’s publication criteria as it currently stands. Therefore, we invite you to submit a revised version of the manuscript that addresses the points raised during the review process.

ACADEMIC EDITOR: All limitations should be highlighted and discussed.

We look forward to receiving your revised manuscript.

Kind regards,

Vincenzo Lionetti, M.D., PhD

Academic Editor

PLOS ONE

Journal Requirements:

Reviewers' comments:

Reviewer's Responses to Questions

**Comments to the Author**

1. If the authors have adequately addressed your comments raised in a previous round of review and you feel that this manuscript is now acceptable for publication, you may indicate that here to bypass the “Comments to the Author” section, enter your conflict of interest statement in the “Confidential to Editor” section, and submit your "Accept" recommendation.

Reviewer #1: (No Response)

Reviewer #2: All comments have been addressed

Reviewer #3: All comments have been addressed

2. Is the manuscript technically sound, and do the data support the conclusions?

Reviewer #1: No

Reviewer #2: Yes

Reviewer #3: Yes

3. Has the statistical analysis been performed appropriately and rigorously? 

Reviewer #1: Yes

Reviewer #2: Yes

Reviewer #3: Yes

4. Have the authors made all data underlying the findings in their manuscript fully available?

Reviewer #1: Yes

Reviewer #2: Yes

Reviewer #3: Yes

5. Is the manuscript presented in an intelligible fashion and written in standard English?

Reviewer #1: Yes

Reviewer #2: Yes

Reviewer #3: No

6. Review Comments to the Author

Reviewer #1: Dear authors, thank you for the revised version of the manuscript. It has definitely improved, however my main objection is that I still do not feel that the data fully support the conclusion that magnesium is "better" than lidocaine. Yes there is a difference between the groups but the choice of study drug was based on anaesthetist's preference, thus other factors could explain the results or at least act as confounders.. There was no difference in MAC but MAC at induction is a highly dynamic parameter and thus difficult to measure and compare. At least these issues should be mentioned in the discussion as limitations.

Reviewer #2: You have addressed the concerns raised from reviewers. Thus as my personal opinion this manuscript is suited for publication in this journal.

Reviewer #3: (No Response)

7. PLOS authors have the option to publish the peer review history of their article (what does this mean?). If published, this will include your full peer review and any attached files.

Reviewer #1: No

Reviewer #2: No

Reviewer #3: No

---

## [Author Response · Author response to Decision Letter 1]

20 Apr 2021

First of all, we would like to acknowledge all reviewers for giving your time to review our manuscript and response letter. Then our responses for reviewers’ comment are put as follow.

Reviewer 1 

Dear reviewer!; we accept your concern regarding confounding factors, but we were thinking in observational studies these confounding factors are default and that is why we did not mention. Now we have incorporated the limitation and relevance of the study in the manuscript. 

The other concern is ‘the choice of the study drug is by the preference of anesthetist’. In this point what we want to say is the nature of the study is observational and we did not enforce anesthetists to change their plan for the purpose of our study. The data collectors just observe what was happening and collecting data. The study was done in resource limited country, so with it’s limitations we think this study can give information for researchers. Therefore, dear reviewer please review our manuscript for the third time thank you. 

Reviewer2

Dear reviewer! Thank you for assuring as we have addressed your comments 

Reviewer 3

Dear reviewer! we have tried to rewrite the manuscript with language experts

---

## [Decision Letter · Decision Letter 2]

4 May 2021

PONE-D-20-36884R2

Comparison of Intravenous Magnesium sulphate and Lidocaine for Attenuation of Cardiovascular Response to Laryngoscopy and Endotracheal Intubation in Elective Surgical Patients at Zewditu Memorial Hospital Addis Ababa, Ethiopia.

PLOS ONE

Dear Dr. Misganaw,

Thank you for submitting your manuscript to PLOS ONE. After careful consideration, we feel that it has merit but does not fully meet PLOS ONE’s publication criteria as it currently stands. Therefore, we invite you to submit a revised version of the manuscript that addresses the points raised during the review process.

ACADEMIC EDITOR: some limitations of the study should be added and discussed

We look forward to receiving your revised manuscript.

Kind regards,

Vincenzo Lionetti, M.D., PhD

Academic Editor

PLOS ONE

Journal Requirements:

Reviewers' comments:

Reviewer's Responses to Questions

**Comments to the Author**

1. If the authors have adequately addressed your comments raised in a previous round of review and you feel that this manuscript is now acceptable for publication, you may indicate that here to bypass the “Comments to the Author” section, enter your conflict of interest statement in the “Confidential to Editor” section, and submit your "Accept" recommendation.

Reviewer #1: (No Response)

Reviewer #3: All comments have been addressed

2. Is the manuscript technically sound, and do the data support the conclusions?

Reviewer #1: No

Reviewer #3: Yes

3. Has the statistical analysis been performed appropriately and rigorously? 

Reviewer #1: Yes

Reviewer #3: N/A

4. Have the authors made all data underlying the findings in their manuscript fully available?

Reviewer #1: Yes

Reviewer #3: Yes

5. Is the manuscript presented in an intelligible fashion and written in standard English?

Reviewer #1: Yes

Reviewer #3: Yes

6. Review Comments to the Author

Reviewer #1: Dear authors, thank you for adding a discussion of some of the limitations in the discussion section and that you recommend a randomized trial. However I still feel that the conclusion that magnesium is better than lidocaine is not justified, based on my previous arguments. Instead one could write that magnesium was associated with a more favorable hemodynamic response. Also the fact that the study drug was selected by the anesthetist involved should be mentioned in the limitations.

Reviewer #3: (No Response)

7. PLOS authors have the option to publish the peer review history of their article (what does this mean?). If published, this will include your full peer review and any attached files.

Reviewer #1: No

Reviewer #3: No

---

## [Author Response · Author response to Decision Letter 2]

7 May 2021

First of all, we would like to acknowledge all reviewers for giving your time to review our manuscript repeatedly. Then we have tried to reflect our responses as follow for reviewers according to concerns.

Reviewer 1 

Dear reviewer! Thank you for your concern to be our article scientifically sounded. And you are not satisfied on our conclusion ‘magnesium is better than lidocaine’. Even though we concluded standing from the statistical analysis we can accept your expression ‘pretreatment of magnesium sulphate is associated with a more favorable hemodynamic response’. For our study design we all are comfortable for this expression and will take it as it is. The other concern is ‘the choice of the study drug is by the preference of anesthetist’. The pushing factor to study on this topic is our clinical observation, anesthetists in our study setting are using these drugs as a prophylaxis but there is a confusion which agent is more preferable than the other. So based on this situation we designed observational cohort to study this topic and we are not sure whether it is limitation or not. But you strongly argued this is the limitation of the study so we have incorporated in the manuscript.

Reviewer 3

Dear reviewer! Thank you for assuring as we have addressed your comments

---

## [Decision Letter · Decision Letter 3]

11 May 2021

PONE-D-20-36884R3

Comparison of Intravenous Magnesium sulphate and Lidocaine for Attenuation of Cardiovascular Response to Laryngoscopy and Endotracheal Intubation in Elective Surgical Patients at Zewditu Memorial Hospital Addis Ababa, Ethiopia.

PLOS ONE

Dear Dr. Misganaw,

Thank you for submitting your manuscript to PLOS ONE. After careful consideration, we feel that it has merit but does not fully meet PLOS ONE’s publication criteria as it currently stands. Therefore, we invite you to submit a revised version of the manuscript that addresses the points raised during the review process.

ACADEMIC EDITOR: please edit the abstract in accord with Reviewer's suggestion

We look forward to receiving your revised manuscript.

Kind regards,

Vincenzo Lionetti, M.D., PhD

Academic Editor

PLOS ONE

Journal Requirements:

Reviewers' comments:

Reviewer's Responses to Questions

**Comments to the Author**

1. If the authors have adequately addressed your comments raised in a previous round of review and you feel that this manuscript is now acceptable for publication, you may indicate that here to bypass the “Comments to the Author” section, enter your conflict of interest statement in the “Confidential to Editor” section, and submit your "Accept" recommendation.

Reviewer #1: (No Response)

2. Is the manuscript technically sound, and do the data support the conclusions?

Reviewer #1: Yes

3. Has the statistical analysis been performed appropriately and rigorously? 

Reviewer #1: Yes

4. Have the authors made all data underlying the findings in their manuscript fully available?

Reviewer #1: Yes

5. Is the manuscript presented in an intelligible fashion and written in standard English?

Reviewer #1: Yes

6. Review Comments to the Author

Reviewer #1: Dear authors, I feel the main text of the manuscript is now technically sound. Only the abstract needs to be revised reflecting the last changes to the main text. At the moment it still reads that magnesium sulphate is better than lignocaine.

7. PLOS authors have the option to publish the peer review history of their article (what does this mean?). If published, this will include your full peer review and any attached files.

Reviewer #1: No

---

## [Author Response · Author response to Decision Letter 3]

11 May 2021

First of all, we would like to acknowledge reviewer1 for giving your time to review our manuscript repeatedly. Then we corrected the manuscript 

Reviewer 1 

Dear reviewer! Thank you so much! For reviewing each components of the manuscript carefully. And we have corrected it.

---

## [Decision Letter · Decision Letter 4]

17 May 2021

Comparison of Intravenous Magnesium sulphate and Lidocaine for Attenuation of Cardiovascular Response to Laryngoscopy and Endotracheal Intubation in Elective Surgical Patients at Zewditu Memorial Hospital Addis Ababa, Ethiopia.

PONE-D-20-36884R4

Dear Dr. Misganaw,

We’re pleased to inform you that your manuscript has been judged scientifically suitable for publication and will be formally accepted for publication once it meets all outstanding technical requirements.

Kind regards,

Vincenzo Lionetti, M.D., PhD

Academic Editor

PLOS ONE

Additional Editor Comments (optional):

Reviewers' comments:

Reviewer's Responses to Questions

**Comments to the Author**

1. If the authors have adequately addressed your comments raised in a previous round of review and you feel that this manuscript is now acceptable for publication, you may indicate that here to bypass the “Comments to the Author” section, enter your conflict of interest statement in the “Confidential to Editor” section, and submit your "Accept" recommendation.

Reviewer #1: All comments have been addressed

2. Is the manuscript technically sound, and do the data support the conclusions?

Reviewer #1: Yes

3. Has the statistical analysis been performed appropriately and rigorously? 

Reviewer #1: Yes

4. Have the authors made all data underlying the findings in their manuscript fully available?

Reviewer #1: Yes

5. Is the manuscript presented in an intelligible fashion and written in standard English?

Reviewer #1: Yes

6. Review Comments to the Author

Reviewer #1: Dear authors, I am fully satisfied with your corrections. Thank you for your perseverance. I also want to congratulate you for your contribution to the science of anesthesia despite having very limited resources.

7. PLOS authors have the option to publish the peer review history of their article (what does this mean?). If published, this will include your full peer review and any attached files.

Reviewer #1: No

---

## [Editor Report · Acceptance letter]

21 May 2021

PONE-D-20-36884R4 

Comparison of Intravenous Magnesium sulphate and Lidocaine for Attenuation of Cardiovascular Response to Laryngoscopy and Endotracheal Intubation in Elective Surgical Patients at Zewditu Memorial Hospital Addis Ababa, Ethiopia. 

Dear Dr. Misganaw:

I'm pleased to inform you that your manuscript has been deemed suitable for publication in PLOS ONE. Congratulations! Your manuscript is now with our production department. 

Kind regards, 

on behalf of

Prof. Vincenzo Lionetti 

Academic Editor

PLOS ONE